# Identification of the EdcR Estrogen-Dependent Repressor in *Caenibius tardaugens* NBRC 16725: Construction of a Cellular Estradiol Biosensor

**DOI:** 10.3390/genes12121846

**Published:** 2021-11-23

**Authors:** Juan Ibero, Beatriz Galán, José L. García

**Affiliations:** Center for Biological Research Margarita Salas, Department of Microbial and Plant Biotechnology, Spanish National Research Council, Ramiro de Maeztu 9, 28040 Madrid, Spain; juanibecab@gmail.com (J.I.); bgalan@cib.csic.es (B.G.)

**Keywords:** *Caenibius tardaugens*, estrogen, 17β-estradiol catabolism, estrogen metabolism, transcriptional regulation, TetR regulator, estrogen biosensor, *Escherichia coli*

## Abstract

In this work, *Caenibius tardaugens* NBRC 16725 (strain ARI-1) (formerly *Novosphingobium tardaugens)* was isolated due to its capacity to mineralize estrogenic endocrine disruptors. Its genome encodes the *edc* genes cluster responsible for the degradation of 17β-estradiol, consisting of two putative operons (OpA and OpB) encoding the enzymes of the upper degradation pathway. Inside the *edc* cluster, we identified the *edcR* gene encoding a TetR-like protein. Genetic studies carried out with *C. tardaugens* mutants demonstrated that EdcR represses the promoters that control the expression of the two operons. These genetic analyses have also shown that 17β-estradiol and estrone, the second intermediate of the degradation pathway, are the true effectors of EdcR. This regulatory system has been heterologously expressed in *Escherichia coli*, foreseeing its use to detect estrogens in environmental samples. Genome comparisons have identified a similar regulatory system in the *edc* cluster of *Altererythrobacter estronivorus* MHB5, suggesting that this regulatory arrangement has been horizontally transferred to other bacteria.

## 1. Introduction

Endocrine disruptors (EDCs) are chemical contaminants that interfere with the endocrine system and produce adverse effects in both humans and wildlife. The exposure to estrogens, in concentrations as low as 1 ng/L, has been reported to cause feminization, decreased expression of secondary sex characteristics and a reduction in egg fertilization in fish and amphibians [1]. Natural estrogens and androgens enter the environment through the excretions of humans, domestic or farm animals and wildlife. The 17β-estradiol (E2) is a ubiquitous pollutant usually found in soil and water systems [2,3,4]. The complete mineralization of estrogens to CO_2_ can be accomplished aerobically [5,6,7,8,9,10,11,12,13,14] or anaerobically [15] by a limited number of bacteria, mainly from the phyla Proteobacteria and Actinobacteria. Different biochemical and genetic studies have assigned function to many catabolic genes involved in estrogen catabolism, but numerous issues concerning the degradative pathway and its regulation remain to be fully elucidated [10,16,17,18,19].

*Caenibius**tardaugens* (formerly *Novosphingobium tardaugens)* is an aerobic bacterium, isolated from a sewage treatment plant in Tokyo due to its capacity to mineralize E2 and other EDCs [5,20]. Taking advantage of the fact that the genome of this bacterium has been recently assembled in a single contig [21], we have used it as a model system to study the degradation pathways that mineralize EDCs [22,23] (Figure 1). 

These studies have allowed us to identify two gene clusters, named *SD* and *edc*, responsible for the degradation of testosterone (TES) and E2, respectively [22,23]. The transcriptomic analysis enabled the characterization of the *edc* cluster that is organized in two divergent operons, OpA (*EGO55_13525–EGO55_13565*) and OpB (*EGO55_13570–EGO55_13600*), and a divergently expressed gene, *EGO55_13520*, which encodes a putative TetR-like transcriptional regulator [23] (Figure 2). The E2 degradation pathway is tightly regulated, reinforcing the idea that the TetR-like protein could play a regulatory role [23]. The TetR family of regulators is a large family of one-component bacterial signal transduction systems and their regulatory mechanisms, as well as their large variety of effectors, have been extensively studied [24,25]. Although some of these regulators have been proposed to be involved in the metabolism of steroid compounds in bacteria, none have been experimentally shown to interact specifically with estrogens (see discussion).

In this work, we demonstrate that EdcR regulates the *edc* cluster in *C. tardaugens*. This regulator acts as a repressor of the expression of the *edc* cluster genes. We determined that E2 and E1 work as effectors to de-repress the system. By heterologously expressing this regulatory system in *Escherichia coli,* we constructed a cell biosensor to detect the presence of E2 and E1 in environmental samples.

## 2. Materials and Methods

### 2.1. Chemicals

Testosterone (TES), 17β-estradiol (E2), estrone (E1), 4-hydroxyestrone (4-OHE1), estriol (E3), ethinylestradiol (EE2), pyruvate (Pyr), chloroform, n-hexane, ethyl acetate, sulphuric acid and acetonitrile were purchased from Sigma (Steinheim, Germany). Randomly methylated β-cyclodextrin (TRMB-T Randomly Methylated BCD) (CDX) was purchased from Cyclodex (Alachua, FL, United States).

### 2.2. Strains and Growth Media

Bacterial strains and plasmids used in this study are listed in Table 1. *C. tardaugens* NBRC 16725 (strain ARI-1) was purchased from the Leibniz-Institut DSMZ type culture collection. Nutrient broth (NB) (Difco^TM^, Burlington, NJ, USA) was used as rich medium to grow this strain at 30 °C in an orbital shaker at 200 rpm. This strain was also cultured in minimal medium M63 [KH_2_PO_4_ (136 g/L), (NH_4_)_2_SO_4_ (20 g/L), FeSO_4_·7H_2_O (5 mg/L), pH 7.0] supplemented with 0.39 mM CaCl_2_, 1 mM MgSO_4_ and different carbon sources. Steroids and pyruvate stock solutions were prepared in PBS buffer and 70 mM CDX to reach a final carbon concentration of 54 mM in 20 mM CDX in the culture medium. *Escherichia coli* DH10B was grown at 37 °C in an orbital shaker at 200 rpm in lysogeny broth (LB) medium [26]. The appropriate antibiotics, chloramphenicol (34 µg/mL), kanamycin (50 µg/mL) or rifampicin (50 µg/mL) were added when needed.

### 2.3. DNA Manipulation

Molecular biology and DNA manipulations were performed as described elsewhere [27]. *C. tardaugens* genomic DNA was extracted as described before [21]. Plasmid DNA was purified using High Pure Plasmid Isolation Kit (Roche, Basel, Switzerland). DNA fragments were purified with QIAquick PCR Purification Kit (Qiagen, Düsseldorf, Germany) or QIAquick Gel Extraction Kit (Qiagen, Düsseldorf, Germany). *E. coli* cells were transformed using the RbCl method or by electroporation (Gene Pulser; Bio-Rad, Hercules, USA) [31]. DNA amplification was performed in a Mastercycler Gradient (Eppendorf, Hamburg, Germany) using the oligonucleotides listed in Appendix A, which were purchased from Sigma (Steinheim, Germany). Phusion High-Fidelity DNA Polymerase (New England Biolabs (NEB), Ipswich, MA, USA) was used for cloning amplifications and *Taq* DNA polymerase (Biotools, Madrid, Spain) for screening and RT-PCR assays. All PCR products were checked by agarose gel electrophoresis and those aimed for cloning were confirmed by DNA sequencing by Secugen S.L. (Madrid, Spain). Digestion of DNA fragments was done using restriction enzymes from NEB and ligation was performed with Instant Sticky-end Ligase Master Mix (NEB). Construction of pSEVA237M*Pb* was performed as follows: first digesting pSEVA237M-BCD2-14g with *Pac*I and *Avr*II restriction enzymes to excise 14 g promoter; then digesting pSEVA237*Pb*-edcA [23] using *Pac*I and *Xba*I restriction enzymes to extract *P_b_* promoter and finally ligating the core of pSEVA237M-BCD2 plasmid with *P_b_* promoter fragment (Appendix A). pSEVA65edcR plasmid was constructed digesting pSEVA23edcR with *Pac*I and *Spe*I restriction enzymes to extract a DNA fragment containing *P_lexA_* promoter and *edcR* gene and ligating it into pSEVA651 plasmid backbone.

### 2.4. RNA Manipulation

Total RNA of *C. tardaugens* cells was extracted from 12 ml of cultures grown in minimal medium with 20 mM CDX and TES or pyruvate as carbon sources. Cells were harvested in mid exponential phase (OD_600_ 0.6) and stored at −80 °C. Pellets were thawed and cells were lysed in 400 µL TE buffer (10 mM Tris-HCl, 1 mM EDTA, pH 7.5) containing lysozyme (50 mg/mL) following three freezing/thawing cycles. High Pure Isolation Kit (Roche, Basel, Switzerland), followed by DNA-free^TM^ DNA Removal Kit (Invitrogen, Waltham, MA, USA) treatment, was used to obtained pure RNA. Purity and concentration were measured in a ND1000 spectrophotometer (Nanodrop Technologies, Thermo Fisher Scientific, Waltham, MA, USA). cDNA used as the template for RT-PCR was obtained with a Transcriptor First Strand cDNA Synthesis Kit (Roche, Basel, Switzerland) following the manufacturer’s instructions, where 1 µg of total RNA was used as the template in 20 µL of reverse transcription reaction. RT-PCR was performed adding 1 µL of cDNA, 0.5 mM of each oligonucleotide, 0.25 mM dNTPs, 5% DMSO (*v/v*) and 1 U of *Taq* DNA polymerase to a reaction mixture of 30 µL in a 20-cycle PCR. Oligonucleotides used in RT-PCR expression analysis are listed in Appendix A. To check that RNA was not contaminated with DNA, a 30-cycle PCR was performed using 200 ng total RNA as template and oligonucleotides 5RTRecAf and 5RTRecAr (Appendix A).

### 2.5. Construction of C. tardaugens Knockout Strains

The knockout strains were constructed by double homologous recombination using the suicide vector pK18mobsacB [28] as described before [22]. *C. tardaugens* genomic DNA was used as template to amplify two fragments of ≈700 bp containing the upstream and downstream regions of the gene to delete UP and DOWN fragments (Table 1 and Appendix A), respectively. The fragments were digested with the appropriate restriction enzymes and cloned in the unique sites of the plasmid. The ligation product was transformed into *E. coli* DH10B competent cells and once recombinant candidates were PCR-checked, the cloned region was confirmed by sequencing. The plasmids were transformed by triparental conjugation [32] into *C. tardaugens* Rf^R^ as recipient strain using *E. coli* HB101 (pRK600) [33] as a helper and *E. coli* DH10B, harboring the corresponding vector, as a donor, as described before [23].

### 2.6. Complementation of C. tardaugens Knockout Strains

The mutant strains were transformed with pSEVA23 plasmids harboring the corresponding deleted genes under the expression control of *P_lexA_* constitutive promoter [29] (Table 1) using triparental conjugation, following the same protocol used to construct *C. tardaugens* knockout strains.

### 2.7. Fluorescence Measurement of Cultures on a Plate Reader

The fluorescence signal of GFP (Green Fluorescent Protein) in cultures of strains carrying transcriptional fusions was measured using a Varioskan Flash microplate reader (Thermo Scientific, Waltham, MA, USA), using excitation and emission wavelengths of 485 nm and 511 nm, respectively. The measurements were made in 96-well plates and three technical replicates of each sample were taken into three different wells. The measurement was performed by loading into each well 200 µL of cells at an OD_600_ of 1.0, previously washed with 0.85% (*w*/*v*) NaCl solution in H_2_Omq. The measurements were normalized to OD and to a control where no GFP (only 0.85% (*w/v*) NaCl solution in H_2_Omq) was used.

### 2.8. In Silico Genomic Analysis

Analysis of nucleotide sequences, chromatograms from sequencing reactions, design of genetic engineering experiments, design and analysis of oligonucleotides and mapping of reads obtained from transcriptome sequencing against the *C. tardaugens* genome, were done with the program Geneious R11.1.5 (https://www.geneious.com, accessed on 25 October 2021). Identification of conserved promoter sequences with was performed using the tools BPROM [34] from the Softberry server (https://www.softberry.com, accessed on 25 October 2021) and SAPPHIRE [35] (https://www.biosapphire.com, accessed on 25 October 2021). Gene product prediction was done with Rapid Annotations using Subsystems Technology (RAST) [36]. Amino acid sequences of individual proteins were compared with those in the databases by using the BLASTp program [37] from the National Center for Biotechnology Information (NCBI; https://blast.ncbi.nlm.nih.gov/Blast.cgi, accessed on 25 October 2021) server. Multiple alignments of nucleotide and amino acid sequences were performed with ClustalOmega [38] from the server of the European Bioinformatics Institute (EMBL-EBI; http://www.ebi.ac.uk/Tools/msa/clustalo/, accessed on 25 October 2021).

## 3. Results

### 3.1. In Silico Analysis of the Main Promoters Responsible for the Expression of C. tardaugens Edc Gene Cluster

Transcriptomic studies have revealed a high induction level of the *edc* cluster genes in the presence of E2 [23]. The structure of the *edc* cluster suggests the existence of two promoters in the intergenic region located between the divergent OpA and OpB putative operons, and at least another promoter upstream the *EGO55_13520* (*edcR*) gene, encoding a putative regulatory protein EdcR that is divergently transcribed to the OpA genes (Figure 2). Furthermore, the large intergenic region of 85 bp between the *EGO55_13595* and *EGO55_13600* genes suggests the presence of another promoter region that can drive the expression of the *EGO55_13600* (*edcT*) gene, encoding a putative transporter (Figure 2).

To get a whole picture of the *edc* gene expression, the raw reads obtained in the RNAseq experiments carried out in the presence of E2 or Pyr were mapped against the genome sequence. As expected, the result reflected a higher number of reads when cells were grown in the presence of E2, which corresponds to the higher level of induction already described (Figure 3). In both conditions, the intergenic region between the *EGO55_13565* and *EGO55_13570* genes (i.e., the initial genes of the OpA and OpB divergent operons, respectively) showed a low number of aligned reads (i.e., an aligning gap) compared with the reads aligned to both catabolic genes, suggesting that this region could contain the two promoters which drive the expression of both operons (Figure 3). A less pronounced aligning gap, but still significant, was observed upstream of the *EGO55_13520* gene suggesting the existence of another promoter region (Figure 3). 

Interestingly, Figure 3 also shows that in the Pyr condition, the number of reads mapped at the *EGO55_13520* (*edcR*) gene was proportionally higher than the average of the reads mapped to the rest of the catabolic genes, suggesting that this gene was more expressed than the other genes of the *edc* cluster in the absence of E2. When the expression of the *edcR* regulatory gene was more precisely compared with the expression of the other genes of *edc* cluster using the number of normalized reads, we observed that the average of normalized reads mapped to *EGO55_13520* gene when cells are cultured on Pyr (1553 transcripts per kilobase million, TPMs) was higher than the reads mapped to OpA (239 TPMs) and OpB (889 TPMs) operons [23]. However, when cells are cultured on E2, *edcR* had slightly less reads per transcription unit (9390 TPMs) than the reads of OpA (14401 TPMs) and OpB (22708 TPMs) operons [23]. This suggests that EdcR might act as a repressor during growth on Pyr, a condition in which the genes of the *edc* cluster are silenced.

It is also worth mentioning that Figure 3 does not show an aligning gap at the end of the OpB operon and the beginning of the *EGO55_13600* (*edcT*) gene, likely suggesting that this gene can be expressed not only by its own promoter, but also by the promoter of the OpB operon.

On the other hand, the transcriptomic analysis revealed a higher induction of the genes of the OpA operon compared to those of the OpB operon, likely due to an overall low expression level in PYR condition of genes of the OpA operon (239 TPMs) compared to those of the OpB operon (889 TPMs) (Figure 3c). Furthermore, there is a significant difference in the expression levels between the *EGO55_13595* gene belonging to the OpB operon and *EGO55_13600* gene, suggesting, as pointed out above, the presence of an additional promoter element driving the expression of the *EGO55_13600* gene (Figure 3a,c). 

Therefore, these analyses support the existence of at least four promoter regions in the *edc* cluster, i.e., *P_r_*, *P_t_, P_a_* and *P_b_*, to control transcription of *edcR* and *edcT* genes, and OpA and OpB operons, respectively (Figure 3).

The promoter regions were analyzed to detect the putative RNA polymerase binding sequences (boxes -10 and -35), as well as the possible operator binding sequences for transcriptional factors using bioinformatic tools (see Materials and Methods). Possible -10 and -35 boxes were identified in these regions, but the boxes of *P_b_* promoter were more similar to the consensus -10 (TATAAT) and -35 (TTGACA) boxes of *E. coli* [39] (Figure 4). On the other hand, a conserved palindromic sequence (TnnCnnTACnAnTnGTAnnGnnA) was identified in the *P_a_* and *P_b_* promoters as a putative operator site. A similar but less conserved sequence was also identified at *P_r_* and *P_t_* ‘promoters. This putative operator sequence is overlapping the -10 box in all these promoters (Figure 4).

When the promoter regions of *C. tardaugens* were compared with the equivalent regions of the *edc* homologous gene clusters of the estrogen-degrading bacteria *Altererythrobacter estronivorus* MHB5 and *Sphingomonas* sp. KC8 [10,14], we observed only highly conserved sequences in the *C. tardaugens* and *A. estronivorus* MHB5 clusters (Figure 4).

### 3.2. Analysis of the EdcR Function

The *edcR* gene (*EGO55_13520*) has been annotated as a transcriptional regulator of the TetR family (Figure 2) according to pfam containing the characteristic domains of the proteins of this family [24,25], i.e., a DNA-binding domain (helix turn helix HTH, PF00440) and a C-terminal domain usually involved in inducer binding and protein oligomerization (domain TetR_C_11, PF16859).

The *edc* cluster of *A. estronivorus* MHB5 contains a *MB02_RS00150* gene encoding a protein that shares 95% amino acid identity at the DNA binding domain (Figure 5) [14]. However, the product of the homologous *KC8_05330* gene from *Sphingomonas* sp. KC8 shows only 53% amino acid identity at the DNA binding domain (Figure 5). This result suggests that the regulator of KC8 might recognize a different operator sequence and supports the exitance of conserved operator sequences in *C. tardaugens* and *A. estronivorus*.

To study the role of EdcR in the metabolism of E2 in *C. tardaugens*, we constructed a ∆edcR deletion mutant that was cultured it in minimal medium supplemented with E2 as the sole carbon and energy source. The ∆edcR strain is able to grow in E2 faster than the wild-type strain (Figure 6a), suggesting that EdcR is not an activator of E2 degradation. When the ∆edcR mutant was complemented with plasmid pSEVA23edcR expressing the gene *edcR* under the control of the *P_lexA_* constitutive promoter (Table 1), the recombinant strain grew slower than the ∆edcR mutant (Figure 6b), suggesting a probable repressor activity of the EdcR protein.

The influence of the absence of *edcR* on the expression of the genes of the *edc* cluster was determined using the ∆edcR mutant strain. Total RNA was extracted from the cells of wild-type and ∆edcR mutant strains growing in minimal medium with E2 or TES as the only carbon and energy sources, and RT-PCRs were performed to determine the relative expression of some representative genes of the *edc* cluster. *EGO55_13555*, *EGO55_13570* and *EGO55_13600* genes were chosen as representative markers of the expression of the OpA and OpB operons. The housekeeping *recA* gene (*EGO55_01665*) was used as internal control. As expected, the expression of the genes was detected in the presence of E2 in the wild-type and ∆edcR mutant strains (Figure 7). However, in the presence of TES the expression was only detected in the ∆edcR mutant strain (Figure 7). This result strongly reinforces the hypothesis that EdcR negatively regulates the expression of these *edc* genes and that the effector molecule could be E2 or some intermediate of the degradation pathway.

### 3.3. Analysis of the EdcR Effectors

As a tool to identify the effector of EdcR, we constructed transcriptional fusions using the *gfp* as a reporter gene in replicative plasmids. The *P_a_* and *P_b_* promoter regions were cloned in the plasmid pSEVA237 (Table 1). The cloned promoter regions comprise approximately 300 bp upstream from the ribosome binding site (RBS) of the *EGO55_13565* and *EGO55_13570* genes, respectively. The resulting plasmids, pSEVA237*Pa* (Appendix A) and pSEVA237*Pb* (Appendix A) were used to transform *C. tardaugens* wild type and ∆edcR mutant strains taking advantage that pSEVA237 is replicative in this strain. The resulting strains C*. tardaugens* (pSEVA237*Pa*), C. tardaugens (pSEVA237*Pb*), *C. tardaugens* ∆edcR (pSEVA237*Pa*) and *C. tardaugens* ∆edcR (pSEVA237*Pb*) were cultivated in NB rich medium supplemented with 2 mM E1 or with CDX as control, and the fluorescence was measured at 36 h. The experiment was carried out using E1 instead of E2, since E1 is the second metabolite of the E2 pathway (Figure 1), and we observed that the strain is also able to grow in E1 as the sole carbon and energy source, suggesting that E1 should act as an effector of the *edc* cluster. The fluorescence intensity in the wild-type strain increased in the presence of E1 but remained constant in the mutant strain (Figure 8). This result correlates with the previous observation that the expression in the wild strain is inducible while, as expected, the expression of *gfp* gene from the *P_b_* and *P_a_* promoters in the ∆edcR mutant is constitutive (Figure 8). This result confirmed that EdcR acts as a transcriptional repressor of OpA and OpB operons in *C. tardaugens* and that E1 or another intermediate metabolite of the pathway is capable of inducing the expression of the *edc* cluster.

To determine if E1 is the true inducer, and to exclude the possible implication of other intermediate metabolites produced further on the degradation pathway, we used the *C. tardaugens* ∆edcA strain, lacking the E1-hydroxylating activity that cannot transform E1 in other metabolites (Figure 1) [23]. It is worth mentioning here that the induction effect of E2 cannot be tested directly in *C. tardaugens* and although E2 is added as an inducer, we are testing the effect of E1, as E2 is transformed by the strain. *C. tardaugens* ∆edcA was cultured in minimal medium supplemented with TES as a carbon and energy source in the presence of E1, and total RNA was extracted. The expression of *EGO55_13555*, *EGO55_13570* and *EGO55_13600* genes was analyzed by RT-PCR. Figure 9 shows that the *edc* genes tested are induced in the presence of E1, suggesting that E1 acts as the true inducer of the *edc* cluster in *C. tardaugens*.

### 3.4. Analysis of the Promoters of the Edc Cluster in E. coli. Construction of an Estrogen Biosensor

To monitor the expression from the *P_a_* and *P_b_* promoters in a heterologous host, we used the pSEVA237*Pa* and pSEVA237*Pb* plasmids carrying the *gfp* gene under the control of *P_a_* and *P_b_* promoters, respectively, able to replicate in *E. coli*. The recombinant strains of *E. coli* DH10B (pSEVA237*PlexA*) (control plasmid), *E. coli* DH10B (pSEVA237*Pa*) and *E. coli* DH10B (pSEVA237*Pb*) were grown on LB and the fluorescence intensity of the cells were analyzed after 5 h. The fluorescence signal was only detected in *E. coli* DH10B (pSEVA237*Pb*), suggesting that only the *P_b_* promoter is recognized by the *E. coli* RNA-polymerase (Appendix A).

To determine if EdcR can be a functional repressor of *P_b_* promoter in *E. coli*, the *edcR* gene was cloned under the control of the *P_lexA_* promoter in the plasmid pSEVA651 (Table 1), generating the plasmid pSEVA65edcR. This plasmid was transformed in *E. coli* DH10B (pSEVA237*Pb*), generating the strain *E. coli* DH10B (pSEVA237*Pb*, pSEVA65edcR). *E. coli* DH10B (pSEVA237*Pb*, pSEVA651) carrying the empty vectors was used as a control. When the strains were grown in LB and the fluorescence intensity was measured after 5 h, we observed that the production of EdcR decreases the fluorescence intensity, suggesting that EdcR acts as a transcriptional repressor on the *P_b_* promoter in *E. coli* (Figure 10).

To confirm that E1 is the true inducer of the *edc* cluster, we tested its effect in *E. coli* DH10B (pSEVA237*Pb*, pSEVA65edcR). A partial recovery of the fluorescence signal was observed in the presence of E1, indicating that E1 is also able to induce the expression of *P_b_* promoter in *E. coli* (Figure 10). Interestingly, E2 had the same effect as E1, suggesting that E2 can be also recognized by EdcR as an inducer (Figure 10). A 2.9- and 2.6-fold increase in signal was observed when E1 and E2 inducers were used, respectively, compared to solvent control. As mentioned above, this induction effect of E2 could not be tested in *C. tardaugens* since a mutant unable to transform E2 into E1 is not available [22].

To test other possible inducers, these experiments were repeated using other steroids such as E3 and 4-OHE1, two natural estrogens that are known to be degraded by *C. tardaugens*; EE2, a synthetic estrogen that is not degraded by this strain; and TES, an androgen that does not induce the *edc* cluster and was used as a negative control. The results are shown in Figure 11 and confirmed that the EdcR only responds to E1 and E2.

Considering that this heterologous regulatory system responds specifically to the presence of estrogens, there is a possibility of developing a biosensor that could be applied to the detection of estrogens in the environment. For this purpose, a new transcriptional fusion of the *P_b_* promoter was constructed using the plasmid pSEVA237M-BCD2-14g (Table 1). This plasmid includes a synthetic bicistronic RBS, BCD2, which increases translation by having two tandem ribosome-binding Shine–Dalgarno motifs [40] and a modified version of GFP, the monomeric superfolder green fluorescent protein, Msf-GFP, whose emission signal is more intense than that of GFP [41]. The synthetic promoter *P_14g_* of plasmid pSEVA237M-BCD2-14g was replaced by the *P_b_* promoter and the resulting plasmid, pSEVA237M*Pb* (Figure 12a), was transformed into *E. coli* DH10B, obtaining the strain *E. coli* DH10B (pSEVA237M*Pb*).

This strain was transformed with the vector pSEVA65edcR, and the resulting strain, *E. coli* DH10B (pSEVA237M*Pb*, pSEVA65edcR), was grown in the presence of E1 and E2. The reporter system responds to the presence of both compounds E1 and E2 (Figure 12c,d), and the signal registered was an order of magnitude higher than that obtained with *E. coli* DH10B (pSEVA237*Pb*, pSEVA65edcR) (Figure 10). The concentration of the inducer molecules E1 and E2 was increased to determine the response curve of the biosensor. The EC50 inferred from the resulting sigmoid functions was 3.32 μM for E2 and 1.38 μM for E1. The differences in the signal fluorescence intensity obtained in the presence or absence of the inducer molecules can be detected by the naked eye without the need for any special equipment (Figure 12e). Thus, the strain generated constitutes a good starting point for the construction of a more sensitive biosensor to determine the presence of estrogens in an environmental sample.

## 4. Discussion

The results presented above demonstrate that the *edcR* gene located contiguously to the *edc* cluster, responsible for E2 degradation, encodes a TetR-like regulator that represses the expression driven by the *P_a_* and *P_b_* promoters. These promoters control the expression OpA and OpB operons encoding the enzymes of the upper E2 degradation pathway. We have also shown that E2 and E1 act as true inducers of EdcR regulator. EdcR is a new member of a very short list of bacterial regulators that have been linked to the control of steroid degradative pathways. The best characterized regulators in this list are the KstR and KstR2 regulators that control the expression of the cholesterol upper and lower degradative pathway, respectively, in actinobacteria [42,43,44,45,46,47,48]. As EdcR, KstR and KstR2 are TetR-like repressors, KstR and KstR2 effector molecules have been identified being 3-oxocholest-4-en-26-oyl-CoA, a metabolite synthetized in the first steps of cholesterol degradation, and HIP-CoA, respectively [46,49,50]. Another example is provided by the regulators involved in TES degradation by *Comamonas testosteroni*, where, according to the published data, the regulation appears to be rather complex. A LuxR-type regulator (TeiR, Testosterone-inducible Regulator), which positively regulates TES degradation, was initially identified in this bacterium [51,52,53]. This positive regulation appears to require a metabolite derived from TES degradation. A *tesR* gene almost identical to *teiR* was also identified in *C. testosteroni* TA441 [54]. TeiR regulator was shown to be a kinase that responds to a variety of different steroids other than TES, and also mediated the induced expression of 3α-hydroxysteroid dehydrogenase/carbonyl reductase involved in TES metabolism, encoded by the *hsdA* gene [55]. Nevertheless, the regulation of *hsdA* expression is even more complex since two repressors, RepA and RepB [56,57], along with HsdR, a LysR-like regulator [58], are also involved. On the other hand, a LuxR-like regulator has been described as a repressor that controls the expression of 3,17β-hydroxysteroid dehydrogenase [59,60]. In this case, TES appears to be the inducer [60]. Other works have identified in *C. testosteroni* a TetR-like regulator, controlling the expression of 3,17β-hydroxysteroid dehydrogenase [61,62,63]. Regarding the regulation of the 4,5-*seco* pathway of steroid degradation, there is only one previous study identifying CrgA and OxyR acting as an activator and a repressor, respectively, of the expression of a 17β-HSDs capable of transforming E2 into E1 in *P. putida* SJTE-1 [19]. HSDs are widely distributed in steroid-degrading bacteria and often present a wide range of substrates, catalyzing reactions in a nonspecific manner [22,64], and the regulation of other HSDs involved in testosterone degradation has been extensively studied [56,57,59,61,62,63,65]. In this regard, the new information gathered in this work becomes of great importance, as it is the first time that the regulation of a cluster essential for estrogen degradation, which contains the specific genes required for the cleavage of the core of the molecule, has been addressed. Furthermore, it is the first time that the participation of a regulator of the TetR family in estrogen metabolism in bacteria has been demonstrated.

In silico analysis of estrogen degradation clusters from different bacteria revealed the existence of proteins homologous to EdcR (*EGO55_13520*) of *C. tardaugens*, encoded by genes *MB02_RS00150*, *F7P65_RS09285*, *KC8_05330* and *IM701_20395* from *A. estronivorus* MH-B5 (Qin et al., 2016), *S. estronivorans* AXB (Qin et al., 2020), *Sphingomonas* sp. KC8 (Chen et al., 2017) and *Novosphingobium* ES2-1 (Li et al., 2020), respectively. Interestingly, the palindromic sequence found overlapping the -10 box in *P_a_* and *P_b_* promoters of *C. tardaugens* is entirely conserved in the *P_a_*- and *P_b_*-equivalent promoter regions of *A. estronivorus*. Negative autoregulation is a common feature of gene regulation [66] and in examples described in steroid metabolism, the palindromic operator sequence has been found in front of the regulatory gene itself, such as in KstR and HsdR [42,65]. The proposed EdcR operator palindrome (TnnCnnTACnAnTnGTAnnGnnA) is only partially conserved in *P_r_*, (TnnCnnTAnnAnTnnTAnnGnnA), suggesting that EdcR would repress its own expression, but likely less efficiently. Although previous studies showed that the identification of repeated sequences does not guarantee that they are the operating regions recognized by the corresponding TetR regulator [25], this approach has been successfully applied in some cases [67]. In addition, HTH DNA-binding domains of EdcR and its homolog in *A. estronivorus* share a high percentage of amino acid sequence identity, which is a characteristic observed, in general, within the TetR family proteins, where the HTH DNA-binding domain is more conserved than the substrate-binding domain [24]. This evidence suggests the existence of a conserved estrogen degradation regulatory system in these two bacteria. However, this conservation cannot be extended to the intergenic regions of *Sphingomonas* sp. KC8 and *Sphingobium estronivorans* AXB, where in silico analysis did not reveal the presence of the palindromic sequence. Other bacteria described as estrogen degrading with genomes available in databases, such as *Achromobacter xylosoxidans* NBRC 15,126 (accession number NZ_BCZG01000023.1) and *Denitratisoma oestradiolicum* DSM 16959(T) (accession number NZ_LR778301.1) do not have a regulatory gene similar to EdcR and have a very different organization of the estrogen degradation cluster than in *C. tardaugens.* Although homologous proteins to EdcR have been found in the genomes of these bacteria, the protein encoded by *AX2_RS23365* in *A. xylosoxidans* NBRC 15,126 presents 79% ID with 30.36% coverage, but with very low statistical significance (2E-05), whereas the proteins encoded by *DENOEST_RS05595* and *DENOEST_RS05530* in *D. oestradiolicum* DSM 16959(T) present 79% and 91% ID with 38.18% and 35.26% coverage and with statistical significance of 2 × 10^−28^ and 4 × 10^−27^, respectively. The *edc* cluster is not present in the estrogen degrading bacteria *Steroidobacter denitrificans* FST, as it metabolizes them under anaerobic conditions, involving a distinct set of genes [68]. There are other bacteria (belonging to different genera) having homologous to EdcR but their estrogen-degrading capacity has not been described.

In this work, we also investigated the expression of the EdcR regulatory system from *C. tardaugens* in a heterologous host such as *E. coli* using the pSEVA plasmid platform. Only the *P_b_* promoter that expresses the genes of OpB operon showed a significant activity in *E. coli*. As the transcriptional fusion of the *P_a_* with the *gfp* gene showed a high activity in *C. tardaugens,* the lack of activity in *E. coli* cannot be ascribed to an incorrect construction. The *P_b_* promoter region shows -35 and -10 boxes more similar to the consensus boxes of *E. coli* than the *P_a_* promoter region that might not allow the *E. coli* RNA polymerase binding. The non-functionality of the *P_a_* promoter in *E. coli* might suggest that a specific transcription factor not produced by *E. coli* is required in *C. tardaugens* to express the genes of OpA operon. The expression levels of different transcription factors were analyzed in the transcriptome of *C. tardaugens* cells growing in E2 [23]; however, none of them showed any induction correlating with that observed in genes of the *edc* cluster.

The construction of the heterologous regulatory system in *E. coli* has allowed identifying E2, in addition to E1, as effectors of EdcR. Due to the high redundancy of 17β-hydroxysteroid dehydrogenase activities in *C. tardaugens* [22], a mutant unable to transform E2 into E1 is not available and thus, it was impossible to distinguish in *C. tardaugens* the induction effect of E2 and E1, since E2 is immediately converted into E1. Moreover, the *E. coli* heterologous systems have opened the possibility to explore them as biosensor chassis to detect estrogen in environmental samples. In this chassis, the mechanisms of TetR-like regulators as well as the wide variety of inducer compounds have been extensively studied [24,25], and several of them have been used to construct biosensors so far [69,70,71,72,73,74,75].

The detection of EDCs is currently carried out by chromatographic analysis [76,77]. Nevertheless, several biosensors have also been developed to detect EDCs. These biosensors are mainly based on antibodies [78,79,80,81,82,83,84,85,86,87,88,89,90,91,92,93,94,95,96,97,98,99,100], aptamers [101,102,103,104,105,106,107,108,109] and the isolated animal estrogen α-receptor (ERα) [93,110,111,112,113,114,115], or integrated in bacteria [116,117,118], fungi [119], yeast [120,121,122], eukaryotic cell lines [123,124,125], plants [126] or transgenic animals [127,128,129,130,131,132,133,134,135,136]. Only a few examples of bacterial biosensors based on specific bacterial regulatory systems useful to detect steroids have been described so far. For instance, taking advantage that the expression of the *hsdA* gene from *C. testosteroni* is regulated by different steroids [137], a *C. testosteroni* mutant harboring a *hsdA*-*gfp* transcriptional fusion has been created to detect the presence of TES, E2 and cholesterol in a competitive manner when compared to other biosensors [138]. More recently, an optical progesterone biosensor was created based on the use of a TetR-like transcription factor named SRTF1 of *Pimelobacter simplex* [139].

In this work, we developed four biosensors based on the use of different recombinant cells: (i) *C. tardaugens* (pSEVA237*Pb*), (ii) *C. tardaugens* (pSEVA237*Pa*), (iii) *E. coli* (pSEVA237*Pb*, pSEVA65edcR) and (iv) *E. coli* (pSEVA237M*Pb*, pSEVA65edcR). The last one allows the detection of the pollutants by the naked eye. Most important is the fact that once the components of the system (regulator, effectors, promoters and operator sequence) are identified, it is possible to improve the sensitivity and the specificity of the system by using different genetic engineering tools. For instance, the EdcR regulator can be engineered to recognize a wider range of effectors or to improve the binding constant the cognate inducers. Promoter and operator sequences can be optimized, and the whole system can be tuned to find the highest sensitivity. EdcR could be used as SRTF1 from *P. simplex* to develop optical biosensors. Additionally, other bacteria different from *E. coli* (e.g., *P. putida*) that could uptake the estrogens more efficiently can be considered as possible chassis to support the biosensor. Finally, the results of this work pave the way to develop new regulatory circuits involving EdcR and specific estrogenic effectors that can be useful for different biotechnological purposes.

## Figures and Tables

**Figure 1 genes-12-01846-f001:**
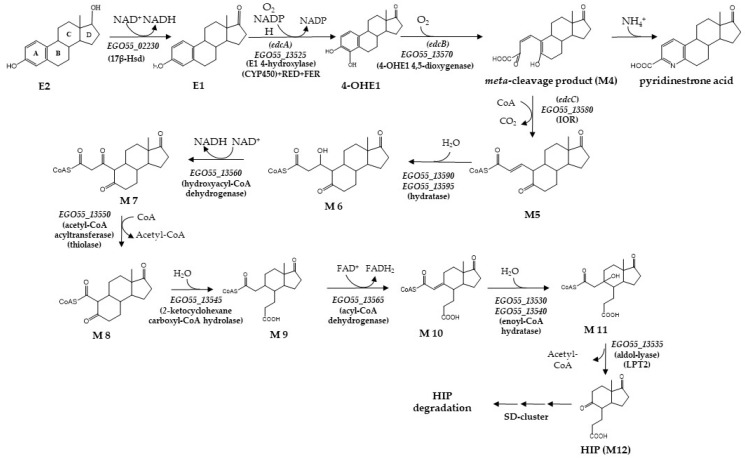
Proposed estrogen degradation pathway in *C. tardaugens* NBRC 16725. Compound names are indicated with an abbreviation: (E2) estradiol; (E1) estrone; (4-OHE1) 4-hydroxystrone; (HIP) 3aα-H -4α(3′-propanoate)-7aβ-methylhexahydro-1,5-indanedione. Enzyme names are: (17β-hsd) 3β, 17β-hydroxysteroid dehydrogenase; (*edcA*) E1 4-hydroxylase; (*edcB*) 4-OHE1 4,5-dioxygenase; (*edcC*) *meta*-cleavage product decarboxylase. The catalytic genes of *C. tardaugens* are indicated in italics, with the nomenclature *EGO55_xxxxx*.

**Figure 2 genes-12-01846-f002:**
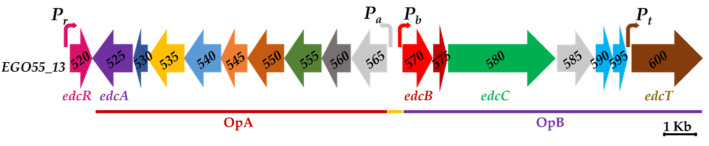
Scheme of the *C. tardaugens* NBRC 16725 estrogen degradation cluster (*edc*). Operons OpA, OpB and the intergenic region are depicted with a line: red, purple and yellow, respectively. Genes annotated as tetR (*EGO55_13520*), cytochrome P450 hydroxylase (*EGO55_13525*), hydratase (*EGO55_13530*), lipid-transfer protein (*EGO55_13535*), enoyl-CoA hydratase/ isomerase (*EGO55_13540*), 2-ketoacyclohexanecarboxyl-CoA (*EGO55_13545*), acetyl-CoA acyltransferase (*EGO55_13550*), hydroxymethylglutaryl-CoA synthase (*EGO55_13555*)*, 3-hydroxyacyl-CoA dehydrogenase* (*EGO55_13560*), acyl-CoA dehydrogenase (*EGO55_13565*), 4-hydroxyestrone-4,5-dioxygenase (*EGO55_13570*), vicinal oxygen chelate containing protein (*EGO55_13575*), indolepyruvate ferredoxin oxidoreductase (*EGO55_13580*), acyl-CoA dehydrogenase (*EGO55_13585*), MaoC dehydratase (*EGO55_13590*), MaoC dehydratase (*EGO55_13595*) and TonB dependent receptor (*EGO55_13600*) are also shown.

**Figure 3 genes-12-01846-f003:**
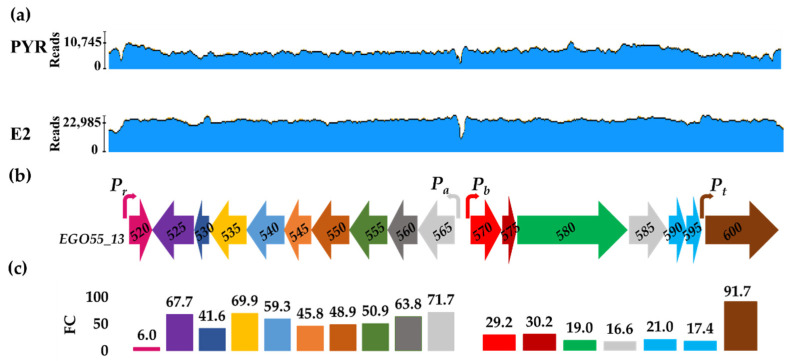
Scheme of gene expression of estrogen degradation cluster (*edc*) in *C. tardaugens* NBRC 16725 cells when grown on E2 compared to PYR condition. (**a**) Mapping of RNA-Seq raw reads against *edc* cluster sequence obtained in PYR and E2 conditions. (**b**) Scheme of *edc* cluster. (**c**) Induction levels of the *edc* cluster measured as fold change (FC).

**Figure 4 genes-12-01846-f004:**
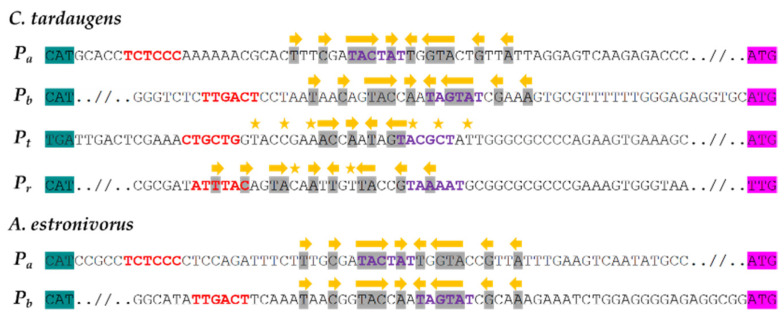
Analysis of promoter sequences of the *edc* cluster. Sequences of the *P_a_*, *P_b_*, *P_t_* and *P_r_* regions of *C. tardaugens* NBRC 16725 and *P_a_* and *P_b_* of *A. estronivorus* MHB5. The -35 and -10 putative boxes are shown in red and purple, respectively. The start codons of the downstream gene are highlighted in pink and those of the gene upstream of the promoter in green, and the palindromic sequences identified are highlighted in gray. Identified palindromes are marked with yellow arrows and nucleotides that differ from the palindrome are marked with yellow stars.

**Figure 5 genes-12-01846-f005:**
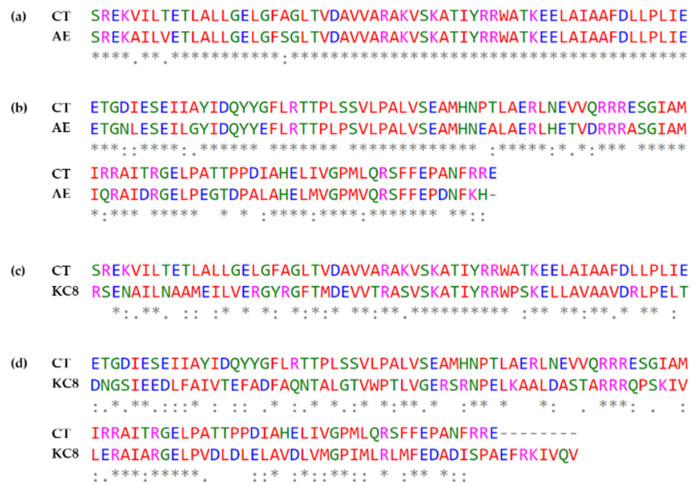
Sequence alignment of the regulators encoded by *EGO55_13520*, *MB02_RS00150* and *KC8_05330*, in *C. tardaugens* NBRC 16725 (CT), *A. estronivorus* MHB5 (AE) and *Sphingomonas* sp. KC8 (KC8), respectively. DNA-binding domains (**a**,**c**) and effector-binding domains (**b**,**d**). Identical and conserved amino acids are marked with "*" and “:”, respectively.

**Figure 6 genes-12-01846-f006:**
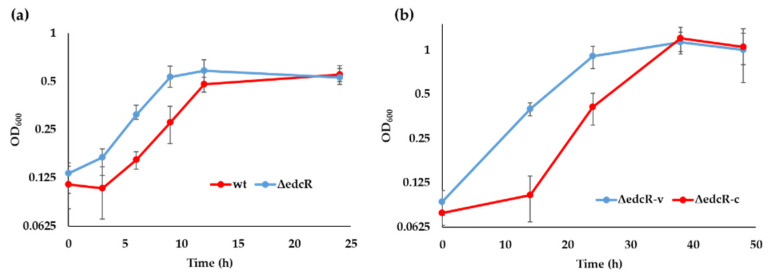
Growth curves (OD_600_) of *C. tardaugens* strains growing in minimal medium supplemented with 2 mM E2. (**a**) *C. tardaugens* wt strain and *C. tardaugens* ΔedcR mutant. (**b**) *C. tardaugens* ΔedcR (pSEVA23*PlexA*) (ΔedcR-v) and *C. tardaugens* ΔedcR (pSEVA23edcR) (ΔedcR-c). The values correspond to the mean of three independent biological replicas (*n* = 3) and error bars correspond to SD.

**Figure 7 genes-12-01846-f007:**
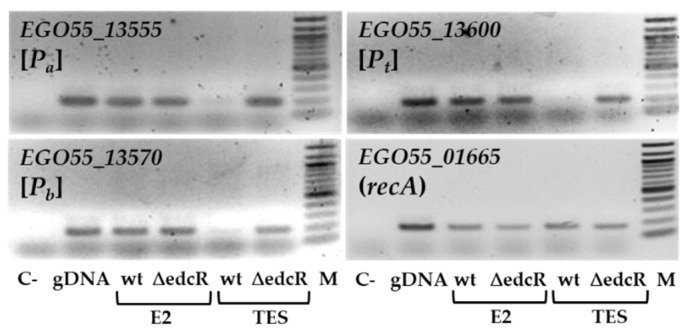
Analysis by RT-PCR of the expression of some genes of the *edc* cluster in *C. tardaugens* wild-type and ∆edcR mutant growing in E2 and testosterone. C- is the control without RNA, gDNA is the control with genomic DNA and M is the 100 bp molecular weight marker.

**Figure 8 genes-12-01846-f008:**
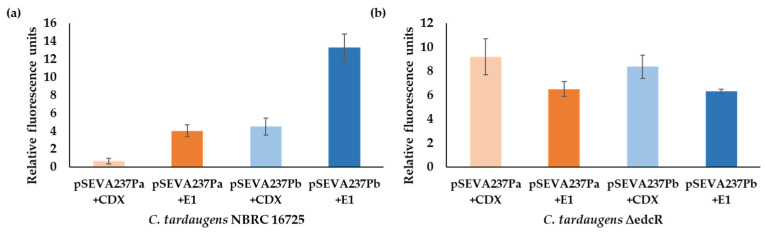
Fluorescence intensity of the transcriptional fusions of the *P_a_* and *P_b_* promoter regions measured in cultures of *C. tardaugens* (**a**) and *C. tardaugens* ∆edcR mutant (**b**) transformed with the plasmids pSEVA237*Pa* or pSEVA237*Pb*, grown in NB carbon and energy source, in the presence of CDX (control) (light colors) or E1 (inducer) (dark colors). The values correspond to the mean of three independent biological replicas (*n* = 3) and error bars correspond to SD.

**Figure 9 genes-12-01846-f009:**
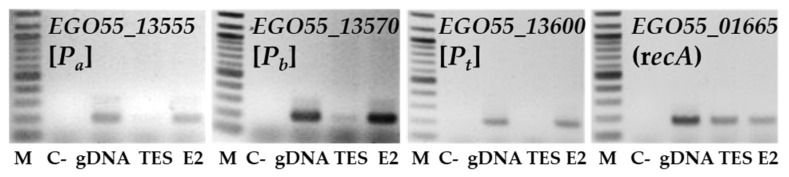
Analysis by RT-PCR of the expression of the genes of the *edc* cluster in *C. tardaugens* ∆edcA mutant growing in testosterone (TES) and testosterone in the presence of E2 (E2). C- is the control without RNA, gDNA is the control with genomic DNA and M is the 100 bp molecular weight marker.

**Figure 10 genes-12-01846-f010:**
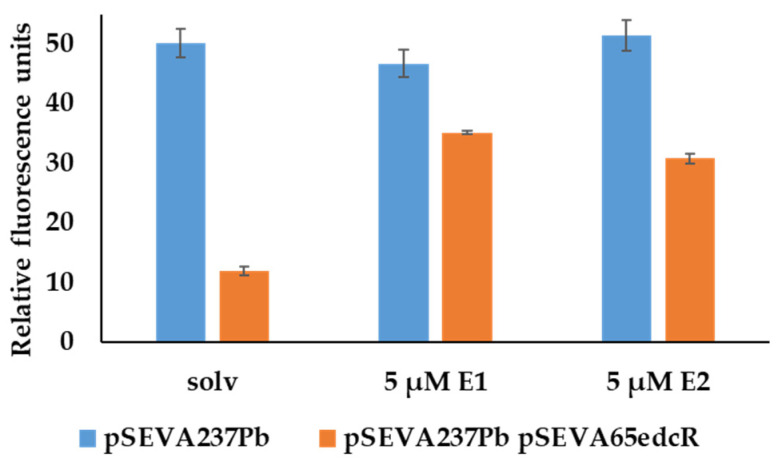
Fluorescence intensity of cultures of *E. coli* DH10B (pSEVA237*Pb*) (pSEVA651) and *E. coli* DH10B (pSEVA237*Pb*) (pSEVA65edcR). Cultures were grown in LB in the presence of the solvent as control (solv) and the inducers, 5 µM E1 or 5 µM E2. The values correspond to the mean of three independent biological replicas (*n* = 3) and error bars correspond to SD.

**Figure 11 genes-12-01846-f011:**
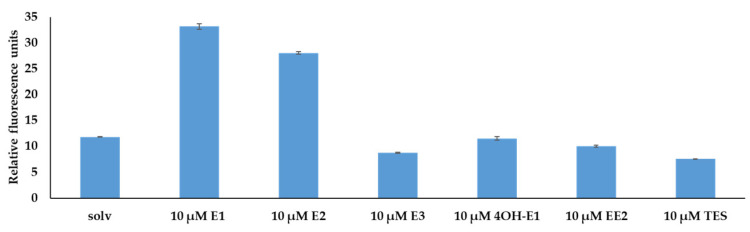
Fluorescence intensity of cultures of *E. coli* DH10B (pSEVA237*Pb*) (pSEVA65edcR) growing in LB in the presence of the solvent as control (solv), 10 µM E1, 10 µM E2, 10 µM E3, 10 µM 4OH-E1, 10 µM EE2 and 10 µM testosterone. The values correspond to the mean of three independent biological replicas (*n* = 3) and error bars correspond to SD.

**Figure 12 genes-12-01846-f012:**
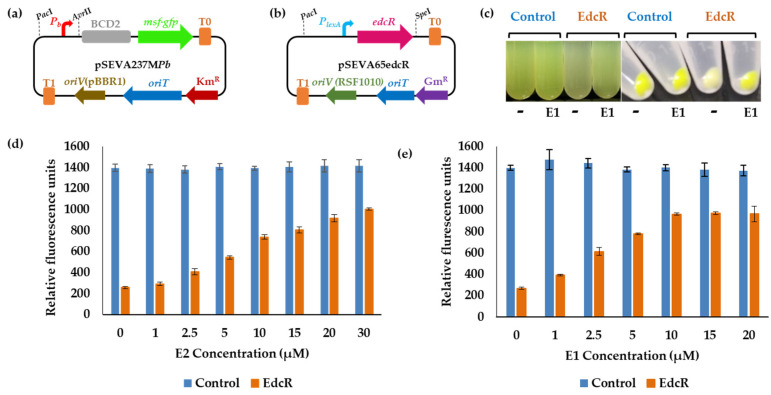
Biosensor systems built using the strains *E. coli* DH10B (pSEVA237M*Pb*) (pSEVA651) and *E. coli* DH10B (pSEVA237M*Pb*) (pSEVA65edcR). Schematic representation of the pSEVA237M*Pb* (**a**) and pSEVA65edcR (**b**) plasmids constructed to develop an estrogen biosensor system. The main elements of the plasmids have been drawn: *P_b_*, *P_b_* promoter region; *P_lexA_*, *P_lexA_* promoter sequence; BCD2, bicistronic RBS; *msf-gfp*, gene encoding the monomeric superfolder green fluorescent protein; *edcR* gene; T0 and T1, transcription terminators; Km^R^, kanamycin resistance gene; Gm^R^, gentamycin resistance gene; *OriT*, transfer origin, *OriV* (pBBR1) and *oriV* (RSF1010), replication origins. The restriction enzymes used for cloning purposes are indicated. (**c**) Visual comparison of the liquid cultures grown in LB in the presence of the solvent as control (-) and 5 µM E1 and the 1.5 ml pellets of the cultures. (**d**,**e**) Fluorescence intensity of the recombinant strains carrying the transcriptional fusion with the *P_b_* promoter. *E. coli* DH10B (pSEVA237M*Pb*) (pSEVA651) control strain is shown in blue and *E. coli* DH10B (pSEVA237M*Pb*) (pSEVA65edcR) (named as EdcR) in orange. The cultures were grown in the presence of the solvent as control (0),1, 2.5, 5, 10, 15 and 20 μM of E1 and 1, 2.5, 5, 10, 15, 20 and 30 μM of E2. The values represented correspond to the mean of three independent biological replicas (*n* = 3) and error bars correspond to SD.

**Table 1 genes-12-01846-t001:** Bacterial strains and plasmids used in this study.

Strains	Genotype and Characteristics	Source/Reference
*C. tardaugens*		
NBRC 16725	wild type strain (ARI-1)	[20]
Rf^R^	Rf^R^ strain efficient for conjugation	[22]
ΔedcA	*C. tardaugens* NBRC 16725 *ΔEGO55_13525*	[23]
ΔedcR	*C. tardaugens* NBRC 16725 *ΔEGO55_13520*	This study
*E. coli*		
DH10B	F^−^, *mcr*A, Δ(*mrr hsdRMS*-*mcrBC*), Φ80d*lacZ*ΔM15, Δ*lacX74*, *deoR*, *recA1*, *araD139*, Δ(ara-leu)7697, *galU*, *galK*, λ^−^, *rpsL*, *endA1*, *nupG*	Invitrogen
HB101	*supE44, ara14, galK2, leuB, lacY1,* ∆(*gpt*-*proA*)62, *rps*L20, *xyl-5, mtl-1, recA13,* ∆(*mcrC*-*mrr*), *hsdS20* (*rB^−^mB^−^*), Sm^R^	[27]
**Plasmids**		
pK18*mob*sacB	Km^R^, ColE *oriV*, Mob+, *lacZα*, *sacB*; vector for allelic exchange homologous recombination mutagenesis	[28]
pK18edcR	pK18*mob*sacB derivative used for *EGO55_13520* deletion	This study
pSEVA237*PlexA*	Km^R^, oriV (pBBR1), constituve expression of *gfp* gene under the control of the *P_lexA_* promoter	[29]
pSEVA23*PlexA*	pSEVA237*PlexA* where *gfp* was deleted using *Xba*I-*Spe*I restriction enzymes and served as empty vector	[23]
pSEVA23edcR	pSEVA237*PlexA* where *gfp* gene was replaced by *EGO55_13520*	This study
pSEVA237*Pa*	pSEVA237*PlexA* where *P_lexA_* promoter was replaced by *P_a_*	This study
pSEVA237*Pb*	pSEVA237*PlexA* where *P_lexA_* promoter was replaced by *P_b_*	This study
pSEVA237*Pt*	pSEVA237*PlexA* where *P_lexA_* promoter was replaced by *P_t_*	This study
pSEVA237M-BCD2-14g	Km^R^, pSEVA237M derivative, synthetic bicistronic RBS BCD2, P14g promoter	Kindly provided by P. Nikel
pSEVA237M*Pb*	Km^R^, pSEVA237M-BCD2-14g derivative where P14g promoter was replaced by *P_b_*	This study
pSEVA651	Gm^R^, *oriV* (RSF1010), standard MCS	[30]
pSEVA65edcR	pSEVA651 containing *EGO55_13520* gene expressed constitutively under *P_lexA_* promoter	This study

Km^R^, kanamycin resistance gene.

## Data Availability

Not applicable.

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
