# Peer review of "Identification of the EdcR Estrogen-Dependent Repressor in Caenibius tardaugens NBRC 16725: Construction of a Cellular Estradiol Biosensor"

_genes, 2021, doi:10.3390/genes12121846_

Round 1
Reviewer 1 Report
The work of Ibero et al., entitled “Identification of the EdcR estrogen-dependent repressor in Caenibius tardaugens NBRC 16725. Construction of a cellular estradiol biosensor”, describes the identification and characterization of a new transcriptional repressor protein EdcR from the bacterium Caenibius tardaugens NBRC 16725 that releases its cognate promoter after binding estrogens. Furthermore, they repurposed this repressor/promoter pair to create an estrogen biosensor, using Escherichia coli. This work is relevant to the field of biosensors because not many are available to detect natural estrogens and androgens that constitute environmental pollutants. As stated, “it is the first time that the participation of a regulator of the TetR family in estrogen metabolism in bacteria has been demonstrated”. Biosensors based on TetR family members are generally easier to optimize (in order to decrease leakiness and/or increase sensitivity), and are promising systems for cell free-based biosensors, emphasizing the importance of this work. Overall, the manuscript is well written, but some parts require attention, namely the paragraph 3.3 (L275-295) where no italics were used, and the final paragraph of the discussion (L522-525), where the authors forgot to delete the instructions (see below for other minor corrections). I would advise the publication of this work after the comments bellow have been addressed, namely the additional experiment to show the response curve of the sensor. Showing a transfer function of e.g. the fluorescence (output signal) vs inducer concentrations (input signal) is standard practice when developing biosensors.
Major comments:
- The authors state that they used the developed biosensor to detect estrogen in environmental samples (Abstract L17-18 and P11, L355-356). Nonetheless, this does not seem to be accurate, since they do not use real samples. The text is misleading to the readers and should be rephrased to indicate that the biosensor was developed foreseeing its use to detect estrogens in environmental samples.
- In section 2.4 the volume of cultures collected for RNA extraction should be stated, as well as the number of cycles used for the RT-PCRs. Regarding the DNA contamination PCR, the primers used for this negative control should be indicated.
- The annotated sequences for the plasmids developed should be available either as the supplementary materials, or deposited in some public database.
- In section 2.7, are the technical replicates three reading of the same well or three wells of the same sample? Please clarify.
- In every graph, the values and errors should be described in the captions, e.g. data represents the mean of three biological replicates (n=3) and error bars correspond to SD (or SEM). In addition, all data should be originated from at least three biological replicates to have statistical meaning.
- The authors should also consider presenting the gel images as negatives, since the contrast is better and the results are easier to visualize.
- Fluorescence levels are described in the Y-axes as “Relative fluorescence units”. Relative to what? The way these calculations were performed should be described in section 2.7.
- When developing biosensors, a transfer function should be presented, depicting fluorescence in function of the inducer(s) concentration, using e.g. serial dilutions. Detection limit and EC50 could then be inferred from the sigmoid. In addition, the levels above which estrogens in environmental samples are considered harmful should be indicated in the text (introduction or discussion).
- In Figures 11 and 13, indicating the fold de-repression compared to the “solv” condition, would help comparing the response efficiencies of E1 vs. E2.
- Comparing Figure 11 to Figure 12, the relative fluorescence using 5 uM of inducer is higher than using 10 uM, while the “solv” condition gives approximately the same value. One would expect that using more inducer would result in higher or similar fluorescence levels. Can the authors elaborate on this?
- I find the discussion too long for the length of the results section, and sometimes seems to wander off the scope of the paper. Please consider shortening it to make the message clearer to the reader.
- L467-469. This is too speculative. Most likely, the difference in Sigma70 factors between the two species explains this difference.
- L518-519. Why is Pseudomonas putida better? Considering the point raised above, would its RNA polymerase recognize Pb?
Minor comments:
L11: Comma after estradiol – “estradiol,”
L12: OpA and OpB between brackets – “(OpA and OpB)”
L38: Comma after Ref 20 – “[20],”
L51: Add reference to Fig. 2 in the text – “[22] (Figure 2).”
L76: “l-” should be “l” and hydration separated by a period mark “FeSO4.7H2O”
L108: Use “reverse transcription” instead of “retrotranscription”
L144: “BPROM” instead of “BPPOM”
L179: “16725” instead of “16,725”
L223: “16725” instead of “16,725”
L251: “slower” instead of “more slowly”
L275-L295: revise the use of italics
L306 and L308: ecd should be in italic
Figure 13a: BCD2 is a regulatory element and should not be represented as an arrow (this way it looks like two genes in an operon). In addition, this figure should also have the schematic representation of pSEVA65edcR for consistency.
L387: “These promoters control the expression” instead of “These promoters control the of expression”
L469-L471: Please add a reference to this statement. The transcriptome was not analyzed in this work.
L508: “four” instead of “fourth”
L522-L525: please delete
Author Response
We will like to thank the reviewe for the valuable suggestions that have been followed to improve the manuscript.
Reviewer 1
Overall, the manuscript is well written, but some parts require attention, namely the paragraph 3.3 (L275-25) where no italics were used, and the final paragraph of the discussion (L522-525), where the authors forgot to delete the instructions (see below for other minor corrections).
This has been corrected in the new version.
I would advise the publication of this work after the comments bellow have been addressed, namely the additional experiment to show the response curve of the sensor. Showing a transfer function of e.g. the fluorescence (output signal) vs inducer concentrations (input signal) is standard practice when developing biosensors.
We have included the experiment suggested by the reviewer in the revised version.
Major comments:
- The authors state that they used the developed biosensor to detect estrogen in environmental samples (Abstract L17-18 and P11, L355-356). Nonetheless, this does not seem to be accurate, since they do not use real samples. The text is misleading to the readers and should be rephrased to indicate that the biosensor was developed foreseeingits use to detect estrogens in environmental samples.
Both paragraphs been rephrased as suggested
- In section 2.4 the volume of cultures collected for RNA extraction should be stated, as well as the number of cycles used for the RT-PCRs. Regarding the DNA contamination PCR, the primers used for this negative control should be indicated.
Done, as suggested
- The annotated sequences for the plasmids developed should be available either as the supplementary materials, or deposited in some public database.
The requested sequences has been included in the supplementary files.
- In section 2.7, are the technical replicates three reading of the same well or three wells of the same sample?
This has been clarified in the text.
- In every graph, the values and errors should be described in the captions, e.g. data represents the mean of three biological replicates (n=3) and error bars correspond to SD (or SEM). In addition, all data should be originated from at least three biological replicates to have statistical meaning.
The sentence “The values correspond to the mean of three independent biological replicas (n=3) and error bars correspond to SD” has been added to the legend of figure 8 and error bars were added in figure 6.
- The authors should also consider presenting the gel images as negatives, since the contrast is better and the results are easier to visualize.
Done, as suggested
- Fluorescence levels are described in the Y-axes as “Relative fluorescence units”. Relative to what? The way these calculations were performed should be described in section 2.7.
A paragraph has been included in this section as suggested.
- When developing biosensors, a transfer function should be presented, depicting fluorescence in function of the inducer(s) concentration, using e.g. serial dilutions. Detection limit and EC50 could then be inferred from the sigmoid.
We have done the experiment as suggested. We have edited figure 12 to include the new data.
In addition, the levels above which estrogens in environmental samples are considered harmful should be indicated in the text (introduction or discussion).
This has been included in the introduction section as suggested.
- In Figures 11 and 13, indicating the fold de-repression compared to the “solv” condition, would help comparing the response efficiencies of E1 vs. E2.
The required data are included in the figure.
- Comparing Figure 11 to Figure 12, the relative fluorescence using 5 uM of inducer is higher than using 10 uM, while the “solv” condition gives approximately the same value. One would expect that using more inducer would result in higher or similar fluorescence levels. Can the authors elaborate on this?
The induction experiments with E1 and E2 have been improved (Figure 12) and they showed that the induction levels are very similar with both compounds.
- I find the discussion too long for the length of the results section, and sometimes seems to wander off the scope of the paper. Please consider shortening it to make the message clearer to the reader.
We have reduced the discussion as suggested.
- L467-469. This is too speculative. Most likely, the difference in Sigma70 factors between the two species explains this difference.
In fact, both things are related. There are 3 genes in C. tardaugens annotated as sigma 70 factors that show only 32-50% identity amino acid sequence to the corresponding sequences in E. coli. Those differences can explain the differences in promoter recognition in both genera.
- L518-519. Why is Pseudomonas putida better? Considering the point raised above, would its RNA polymerase recognize Pb?
The suggestion is based on the idea that the detection limits of the biosensor also depend on the ability of cells to uptake estrogens more efficiently. Therefore, bacteria such as Pseudomonas, which have been reported to metabolize steroids, could probably uptake steroids better than E. coli.
Minor comments:
L11: Comma after estradiol – “estradiol,” Done
L12: OpA and OpB between brackets – “(OpA and OpB)” Done
L38: Comma after Ref 20 – “[20],” Done
L51: Add reference to Fig. 2 in the text – “[22] (Figure 2).” Done
L76: “l-” should be “l” and hydration separated by a period mark “FeSO4.7H2O” Done
L108: Use “reverse transcription” instead of “retrotranscription” Done
L144: “BPROM” instead of “BPPOM” Done
L179: “16725” instead of “16,725” Done
L223: “16725” instead of “16,725” Done
L251: “slower” instead of “more slowly” Done
L275-L295: revise the use of italics Done
L306 and L308: ecd should be in italic Done
Figure 13a: BCD2 is a regulatory element and should not be represented as an arrow (this way it looks like two genes in an operon). In addition, this figure should also have the schematic representation of pSEVA65edcR for consistency. We have added the plasmid map to the figure as suggested
L387: “These promoters control the expression” instead of “These promoters control the of expression” Done
L469-L471: Please add a reference to this statement. The transcriptome was not analyzed in this work. Done
L508: “four” instead of “fourth” Done
L522-L525: please delete Done
Reviewer 2 Report
The authors have convincingly identified a TetR-like regulator in an Estrogen degradation pathway in Caenibius tardaugens NBRC. They have demonstrated that it is sensitive to E2 and E1 and that the system can work heterologously in E. coli.
Points to be addressed:
- Demonstrate that the biosensor can be used in a real-world context.
- The paper notes several times that a mutant that accumulates E2, unable to convert it to E1, is unavailable. Could a knockout of EGO55_02230 be generated?
- I would like to see an experiment confirming interaction between the TetR protein and the promoter sequences. An EMSA with a GFP-tagged TetR and promoter sequences vs non-specific DNA would be a valuable assay
- Inconsistencies in italicization and some typos. Many of the results sections could be briefer.
The figures throughout the paper need improving.
- Remove boxes from fig 1.
- Fig 2. Show the promoters. The figure could be tidied by omitting the gene product names or moving them to the legend.
- When discussing fig 3, quantify the fold change. The axis on 3a is too small to read.
- Fig 6, please show error.
- Fig 7, Nice result, but please show quantitative analysis of RT-qPCR results. Use the CT values from the qPCR run. Normalise to recA and graph.
- Fig 8. Is the fluorescence normalized to OD? What do the error bars represent? Are the data normalized to a no GFP control?
- Fig 9, same as Fig 7. Please remind the reader than E2 is converted directly into E1, because the text talks about E1 and the figure only mentions E2.
- Results 3.4, please remind the reader that the plasmid carries gfp
- Fig 11. What do the error bars represent (SE, SD)? Move the 5 μM to the legend. Could you replace the long plasmid names with more informative ones about what they carry?
- Fig 13. Please graph original system (fig 11) alongside to demonstrate the increase in intensity. Simplify the plasmid labels again. Discuss the fold change relative to the solvent (E2 induction is about 2 fold increased, could this be seen by the naked eye?)
Author Response
We will like to thank the reviewer for the valuable suggestions that have been followed to improve the manuscript.
- Demonstrate that the biosensor can be used in a real-world context. This is an issue that have been addressed by reviewer 1 as well.
We have reformulated the sentences.
- The paper notes several times that a mutant that accumulates E2, unable to convert it to E1, is unavailable. Could a knockout of EGO55_02230 be generated?
Yes, it could be generated but there are 16 homologs in the C. tardaugens genome that should be deleted. As there are other ways to get to the same point (like the heterologous expression) we discard this possibility.
- I would like to see an experiment confirming interaction between the TetR protein and the promoter sequences. An EMSA with a GFP-tagged TetR and promoter sequences vs non-specific DNA would be a valuable assay.
We agree that to have the in vitro confirmation of the EdcR binding to the cognate promoters would be very interesting. However, deciphering the mechanism of the edc cluster regulation at molecular level was not the original aim of this work. In fact, the molecular characterization of the regulator is in progress and will be the matter of a different manuscript
- Inconsistencies in italicization and some typos.
This have been corrected as suggested
- Many of the results sections could be briefer.
We have tried to delete some redundant sentences
- Remove boxes from fig
Done
- Fig 2. Show the promoters. The figure could be tidied by omitting the gene product names or moving them to the legend.
Done.
- When discussing fig 3, quantify the fold change. The axis on 3a is too small to read.
Done
- Fig 6, please show error.
Done
- Fig 7, Nice result, but please show quantitative analysis of RT-qPCR results. Use the CT values from the qPCR run. Normalise to recA and graph.
These experiments show the results of RT-PCR. We have not performed RT-qPCR experiments on these samples as we do not attempt to quantify the precise amount of mRNA, but only show the qualitative differences.
- Fig 8. Is the fluorescence normalized to OD? What do the error bars represent? Are the data normalized to a no GFP control?
We have added a paragraph to answer the reviewer requirements in the experimental section data.
- Fig 9, same as Fig 7. Please remind the reader than E2 is converted directly into E1, because the text talks about E1 and the figure only mentions E2.
We have added a sentence to clarify this point
- Results 3.4, please remind the reader that the plasmid carries gfp.
Done
- Fig 11. What do the error bars represent (SE, SD)?
Error bars correspond to SD. We have added this information when needed.
- Move the 5 μM to the legend. Could you replace the long plasmid names with more informative ones about what they carry?
The plasmid names are long trying to inform better the users and we would like to maintain these names, since they are the names annotated in our lab books. Nevertheless, if the referee believe that it is critical we will change the plasmid names.
- Fig 13. Please graph original system (fig 11) alongside to demonstrate the increase in intensity.
The information is included in the figures.
- Simplify the plasmid labels again. Discuss the fold change relative to the solvent (E2 induction is about 2 fold increased, could this be seen by the naked eye?).
Yes, the difference with E2 can be seen by naked eye as well.
Round 2
Reviewer 1 Report
The authors have addressed all my comments/concerns.
A few more comments:
L27 – change “ng/L” to “ng/l”
L106 – Make paragraph before section 2.4
Figure 12 – replace "TetR" by "EdcR" in panels c, d and e. It is odd and confusing to see TetR in this figure and nowhere else in the manuscript); change the figure caption accordingly - “(named as EdcR)”. Also, change the Y-axis label to “Relative fluorescence units”, instead of "absorbance".
After these minor changes, I support the acceptance of this manuscript for publication.
Reviewer 2 Report
Changes to the text are adequate.